

# Novel sampling methods for monitoring *Anopheles arabiensis* from Eritrea

Jacques D. Charlwood[1,2], Amanuel Kidane Andegiorgish[1,3], Yonatan Estifanos Asfaha[1], Liya Tekle Weldu[1], Feven Petros[1], Lidia Legese[1], Robel Afewerki[1], Selam Mihreteab[4], Corey LeClair[5] and Ayubo Kampango[6,7]

[1] Epidemiology and Public Health, College of Health Sciences, Asmara, Eritrea
[2] Global Health and Tropical Medicine, Instituto de Higiene e Medicina Tropical, Lisbon, Portugal
[3] Department of Epidemiology and Biostatistics, Xi'an Jiaotong University, Shaanxi, China
[4] Malaria Control Program, Asmara, Eritrea
[5] Medicins sans Frontières, Bruxelles, Belgium
[6] Instituto Nacional de Saude, Maputo, Mozambique
[7] Department of Zoology and Entomology, University of Pretoria, Pretoria, South Africa

Corresponding author
Jacques D. Charlwood,
jdcharlwood@gmail.com

## ABSTRACT

**Background:** Studies comparing novel collection methods for host seeking and resting mosquitoes *A. arabiensis* were undertaken in a village in Eritrea. Techniques included an odor baited trap, a novel tent-trap, human landing collection and three methods of resting collection. A technique for the collection of mosquitoes exiting vegetation is also described. Pre-gravid rates were determined by dissection of host seeking insects and post-prandial egg development among insects collected resting.

**Results:** Overall 5,382 host-seeking, 2,296 resting and 357 *A. arabiensis* exiting vegetation were collected. The Furvela tent-trap was the most efficient, risk-free method for the collection of outdoor host-seeking insects, whilst the Suna trap was the least effective method. Mechanical aspirators (the CDC backpack or the Prokopack aspirator) were superior to manual aspiration in a dark shelter but there was no advantage over manual aspiration in a well-lit one. An estimated two-thirds of newly-emerged mosquitoes went through a pre-gravid phase, feeding twice before producing eggs. Mosquitoes completed gonotrophic development in a dark shelter but left a well-lit shelter soon after feeding. One blood-fed female marked in the village was recaptured 2 days after release exiting vegetation close to the oviposition site and another, shortly after oviposition, attempting to feed on a human host 3 days after release. Exit rates of males from vegetation peaked 3 min after the initial male had left. Unfed and gravid females exited approximately 6 min after the first males.

**Conclusions:** Furvela tent-traps are suitable for the collection of outdoor biting *A. arabiensis* in Eritrea whilst the Prokopack sampler is the method of choice for the collection of resting insects. Constructing well-lit, rather than dark, animal shelters, may encourage otherwise endophilic mosquitoes to leave and so reduce their survival and hence their vectorial capacity.

## INTRODUCTION

Efforts to eliminate malaria have increased in recent years and a number of countries are approaching a situation in which local cases of the disease no longer occur (*World Health Organization, 2017a*). To enter the elimination-phase the malaria burden should be reduced to an incidence rate of less than one per 1,000 persons at risk (*World Health Organization, 2018*). In Eritrea the malaria burden has declined from 110 cases/1,000 in 1998 to 6 cases/1,000 in 2017 (*World Health Organization, 2018*), thus, approaching elimination. In order to reach the target of elimination, however, residual malaria transmission must be addressed. This includes monitoring of potential vectors and their control by application of appropriate interventions in existing and newly-active foci (*Killeen, 2014*).

Given the substantial geographic heterogeneities in malaria burden, in Eritrea and elsewhere, assessment of the bionomics of local anophelines is likely to require a range of bespoke interventions for different populations. *Anopheles arabiensis* is the most common vector in east Africa, to date it is the only recorded vector in Eritrea (*Shililu et al., 2004*; *Wiebe et al., 2017*), and is noted for displaying ecological and behavioural plasticity such as readily biting animals or humans, either indoors or outdoors (*Sinka et al., 2010*). As such, control is challenging and its importance as a vector is likely to increase in the future.

In 2017 larvae of *An. arabiensis* were found, on one occasion, on the outskirts of Asmara, the capital, 2,200 m above sea level in the Eritrean Highlands (J.D. Charlwood, 2017, unpublished data). At lower altitudes on the Escarpment mosquito populations are associated with streams, which exit from small-scale dams used to create reservoirs that supply nearby villages with water during the long dry season. Dams and streams are separated by several kilometres and, with typically limited dispersal, mosquitoes are likely to occur in a metapopulation-patchwork of semi-isolated sub-populations, whose ecology is defined as much by the environment as by intrinsic characteristics (*Verdonschot & Besse-Lototoskaya, 2014*). A knowledge of mosquito bionomics in such populations might help in the design of novel control techniques, especially those based on environmental management rather than insecticides.

Assessing outdoor mosquito activity is especially important because outdoor transmission may maintain residual malaria once the major conventional techniques, indoor residual spraying with insecticide (IRS) and use of LLINs, have been deployed (*Killeen et al., 2013*).

Traditionally the 'gold standard' for sampling outdoor biting mosquitoes has been the human landing catch (HLC) in which collectors acting as bait catch mosquitoes attempting to feed on their exposed lower legs (*Silver, 2008*). Whilst this technique is likely to be the sample that best reflects biting exposure outdoors there are a number of problems inherent to it. Human landing catches require a considerable amount of supervision, are expensive to run, and, most significantly, they often expose the collector to pathogen transmission (*Achee et al., 2015*). These considerations have recently led the World Health

Organization (WHO) to prioritize the search for substitutes to HLC (*World Health Organisation, 2017b*) and whilst a number of alternatives have been suggested, 'tent-traps', which catch blood-seeking mosquitoes, prior to biting appear particularly promising (*Charlwood et al., 2017*). Collectors are likely to differ in their efficiency, so that numbers caught may be independent of the actual number biting. Humans produce a large range of volatile chemicals (*Penn et al., 2007*) and vary in their attractiveness to mosquitoes. Individual humans not only attract different numbers of mosquitoes but also different species (*Knols et al., 1995*) so that it becomes difficult to extrapolate from the numbers collected to exposure to disease. These difficulties are common to all techniques that involve humans as baits so alternatives that do not require human involvement may be the most suitable sample. One such sampling technique is the recently developed Suna trap which was both an effective sampling device and control tool in Kenya (*Homan et al., 2016*).

Outdoor biting insects may, nevertheless, rest inside houses or sheds either before or after feeding, providing a target for control if their location can be identified (*Killeen et al., 2016*). Suitable sampling methods for resting mosquitoes are, therefore, another priority. This may be by collection of mosquitoes from houses, sheds or vegetation, e.g. using mechanical or manual aspirators when the insects are resting during the day or when they leave their resting sites at dusk.

A study designed to compare different collection techniques was, therefore, undertaken in September–October 2016 in a village on the escarpment of the Eritrean Highlands. Two types of tent-trap were compared with an odour baited trap (the SUNA trap (*Homan et al., 2016*)) and HLC for the collection of mosquitoes' host-seeking outdoors whilst the CDC backpack aspirator (*Clark, Seda & Gubler, 1994*) and the Prokopack aspirator (*Vazquez-Prokopec, 2009*) were compared to manual aspiration for the collection of those resting indoors. In addition, a novel method for the sampling of insects leaving vegetation at dusk was developed.

# METHODS

## Sample sites

The study took place between the 7th and 23rd of October 2016 in the environs of the village of Adi Boskal (15° 41′ 41.67″ N 38° 38′ 54.59″ E) at an altitude of 1,536 m above sea level in Zoba Anseba, Eritrea (Fig. 1). The village comprises 25 stone-walled houses and is located on a steep hillside 70 m above a field that abutted a stream; the outlet from the local micro-dam. At the time of the survey no rain was recorded and the stream, in which large numbers of *Anopheles* larvae were observed, was very slow flowing with filamentous algae. In addition to the houses there were two animal sheds in the village. At night five goats and a calf were kept in the first shed (shed 1: 3.2 m length × 1.8 m width × 1.8 m height) and 10 goats and five sheep were kept in the other (shed 2: 3.8 m × 3.1 m × 1.6–1.8 m). Close to shed 1, four cows and a donkey were kept in an open-sided, thatched roof corral at night.

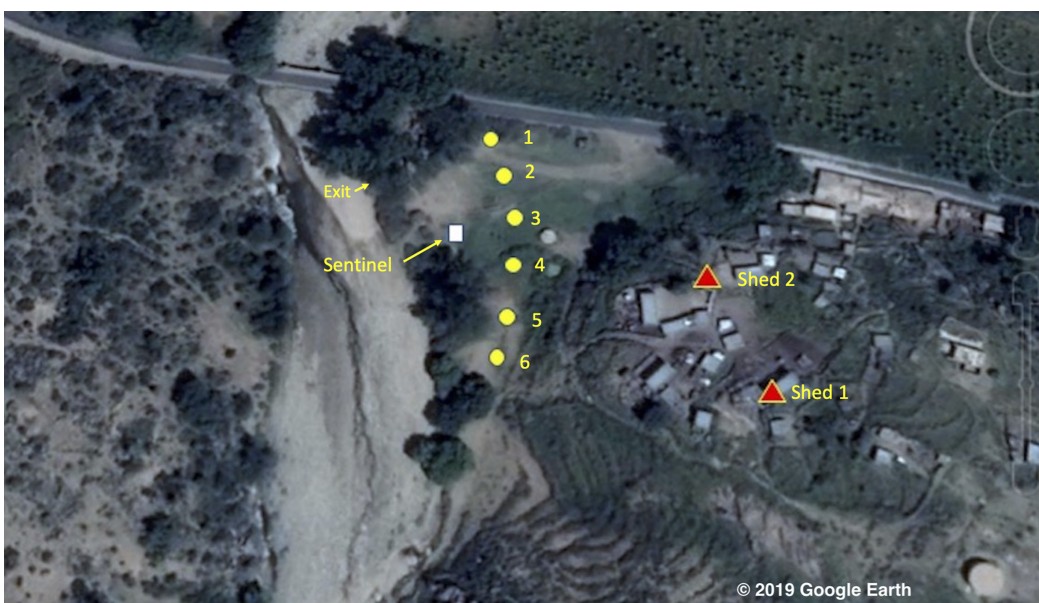

**Figure 1** **Map of the study area © 2019 Google Image Landsat/Copernicus.** The village of Adi Boskal, Anseba Zoba, Eritrea, showing the location of the outdoor trapping sites (yellow circles with numbers representing catch number), the sentinel tent-trap (white square) and the two sheds used for the collection of resting insects (red triangles). Also shown is the location where the exit collections took place. © 2019 Google Image Landsat/Copernicus.               

At other times of the year mosquito numbers are generally very low and in the 12 months prior to the study there had been no cases of malaria from the village reported at the local hospital in the nearby town of Elaberid.

## Study design

A comparison between three methods of collecting post prandial resting mosquitoes were conducted in the two animal sheds shown as red triangles in Fig. 1 and five methods of collecting host-seeking mosquitoes across a transect, shown as yellow circles in Fig. 1, were undertaken. To evaluate the performance of the different types of collection method a randomised block experimental design was applied. As such, collection methods were randomly assigned to sites (blocks) situated approximately 45 m apart to avoid possible interference between collection methods. Then, collectors were also randomly assigned to these locations to perform either HLC catches or act as bait in tent-traps. Collection methods were rotated between sites to account for any possible environmental heterogeneity on trap performance. Accordingly, collectors were rotated between collection methods to reduce the influence of differential attractiveness between collectors on mosquito catch. At each site collection were performed for 5 replicate-days.

A sample of mosquitoes exiting vegetation by the edge of the river valley below the village and a sample from swarms observed over the river bed were also collected at sunset.

## Mosquito collection

### Resting mosquitoes

Mosquitoes resting in the animal sheds were collected either with (1) a CDC-backpack aspirator (*Clark, Seda & Gubler, 1994*), (2) a home-made Prokopack aspirator (*Vazquez-Prokopec, 2009*) or (3) by manual aspiration. The Supplemental File 2 'Sampling routine' indicates the dates when the different collection techniques were used in each shed. Removal sampling (*Southwood, 1978*) was undertaken in order to determine if the different collection methods produced similar population estimates.

Light levels inside the two sheds were measured using a Hand-held LX1010B light-meter (Yingxinguang, Guandong, China) that registered down to 1 Lux.

### Host seeking mosquitoes

The efficiency of the Suna trap was compared to both tent-traps and HLC in the field below the village, shown in Fig. 1. The Supplemental File 'Sampling routine' indicates the location on the transect and date when the different collection techniques were used. At the same time a sentinel tent-trap (with two doors and two traps attached) was run on a nightly basis in one corner of the field (shown as the white square in Fig. 1).

## Landing collections

Human landing collections (HLC) were performed by two teams of three people, in the field below the village, with each individual working a three or 4-h shift (19:00–23:00; 23:00–03:00 and 03:00–06:00). Collectors, using a torch and an aspirator, caught mosquitoes as they attempted to feed on their exposed lower legs and feet. Collected mosquitoes were separated into 4-hourly groups. When not working the collectors acted as bait in tent-traps. Thus, the collectors from the earliest shift replaced the sleepers in the tents who worked the second shift and they themselves replaced the sleepers in the other tents who undertook the last shift of landing collection. The period and location during which the individual collectors worked was alternated on different nights to reduce the influence of differential attractiveness on mosquito catch.

### Furvela tent-trap Type I

Furvela tent-traps (*Charlwood et al., 2017*) were also used for the collection of outdoor biting insects. Odour and exhaled gases from a host leave the tent through an approximately 8 cm opening, equivalent to the diameter of a CDC light-trap, in the door of the tent. A CDC trap (without the light, lid or grid) is placed outside the tent, horizontally, 2 to 3 cm from this opening. On approach to the opening the insects are sucked into the trap and held in a conical collection bag.

In the second part of the study the BG-Lure (a synthetic lure), which, consisted of a mixture of ammonia, L-lactic acid, and caproic acid, in undeclared proprietary concentrations (BioGents HmGb, Regensburg, Germany) designed to mimic human odour to attract mosquitoes (*Homan et al., 2016*), was attached to the outside opening of the tent just below the CDC trap (Fig. 2). This facilitated the dispersion of volatiles from the lure. Collectors slept for two sequential nights in the tents during this phase of the experiment. On one night,

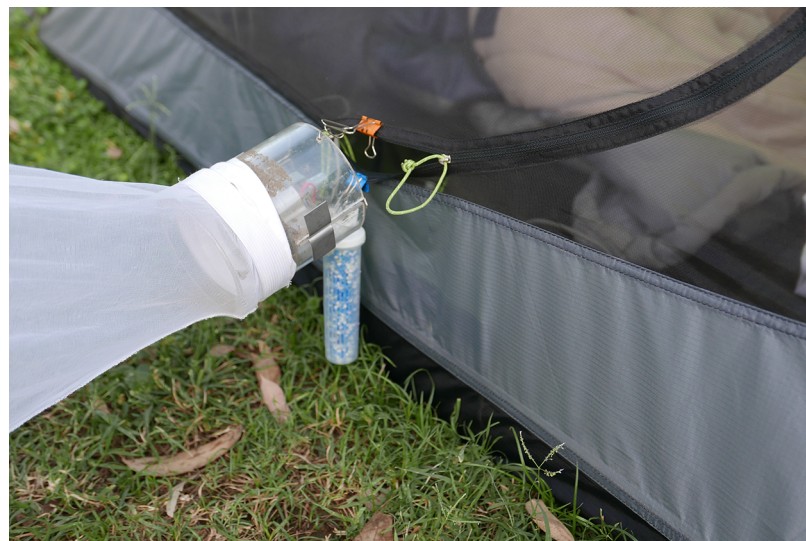

**Figure 2 Furvela tent-trap with lure attached.** Position of the lure attached below the opening of the Furvela tent-trap. Photo © JD Charlwood.

the lure was in use and on the other there was no lure included. Tents were taken down and re-erected every day.

### Furvela tent-trap Type II

A further comparison included a novel design for the attachment of the CDC trap to the tent. In this case an A4 sized plywood board with an 8 cm diameter hole (the size of the internal diameter of the CDC trap) was placed inside the tent at the opening. Two L-shaped clips were used to attach the board to the trap with wing nuts. The clips were positioned at right angles to the tent zip. In this way the size of the opening and the distance of the trap from the tent were more easily standardized than in the basic tent-trap (Fig. 3). A video describing the setting up of the trap is available at https://youtu.be/3UCOhfPGgiw.

### Suna trap

The Suna trap (Biogents, Germany) (Fig. 4) is a conical trap (52 cm high × 39 cm diameter) that has been used to control malaria vectors on Rusinga Island, western Kenya where it reduced malaria transmitted by *An. funestus* but was less effective against *An. gambiae* s.l. (*Homan et al., 2016*). A 12 v battery drives a fan that sucks mosquitoes up through a tube, with a 10 cm diameter opening, into a collection bag. Netting can be placed between the tube and the fan so that collected mosquitoes are not damaged when caught. A non-return gate that is activated when the fan is switched off, or when the tube is removed, means that under these circumstances, the tube acts as a collection cage. The remainder of the base of the trap is perforated with numerous small holes through which the odour from a BG-Lure that is placed inside the trap is blown. There is also the possibility of adding carbon-dioxide to the trap via an external source connected with a tube to an outlet in the trap. In the trial in Rusinga Island 2-butanone was included as a substitute for carbon dioxide (*Homan et al., 2016*). The lack of an effect against

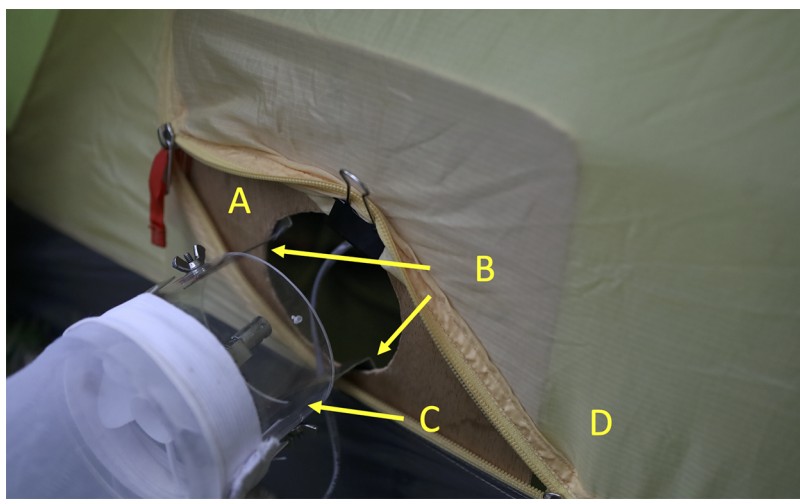

**Figure 3 Furvela tent-trap type II.** Close up of the attachment of the CDC trap body to the tent type II trap. (A) Plywood board with circular opening equivalent to the size of the CDC trap, (B) L-shaped supports for the CDC trap, (C) CDC trap body with light and grid removed and (D) tent with zip open to allow odor and carbon dioxide easy egress. Photo © JD Charlwood.

*An. gambiae* s.l. may have been due to the absence of carbon dioxide in the attractive mix. In the present experiment two traps were used. In one trap carbon dioxide was generated using a sugar and yeast mixture in 2 L of water in a 5 L plastic bottle (*Smallegange et al., 2010*) and in the other carbon dioxide was not used.

The traps were suspended 70 cm above the ground using metal tripods, with the funnel opening set 30cm above the ground (Fig. 4).

### Exit collections from vegetation

Males and recently-emerged virgin females leave their diurnal resting site at dusk to mate, and gravid females do so to oviposit. Endophilic mosquitoes can be collected at this time by placing a netting barrier over the open door of houses (*Charlwood, 2011*); a technique that was adapted here for the collection of mosquitoes leaving their outdoor resting sites. In this case a double-sized (1.8 × 2.2 m), rectangular mosquito net was mounted horizontally in front of an area in the vegetation subjectively considered to be darker than the other vegetation bordering the stream, and the edges sealed with sheets. A collector sat inside the net and caught mosquitoes as they emerged (Fig. 5) Collections were separated into 3-min intervals and light levels were recorded every minute. At the same time swarming insects were observed by looking towards the lightest part of the sky over the dry river bed in areas close to the exit site.

### Mosquito processing

Insects from all collections were sexed, categorized to species or species group using the keys of *Gillies & De Meillon (1968)* and *Gillies & Coetzee (1987)* and females sorted according to their abdominal condition into unfed, part-fed, engorged, semi-gravid and gravid according to Fig. 6. The abdomens of a sample of blood-fed mosquitoes from the resting collections were squashed on filter paper and are available on request. A subsample of

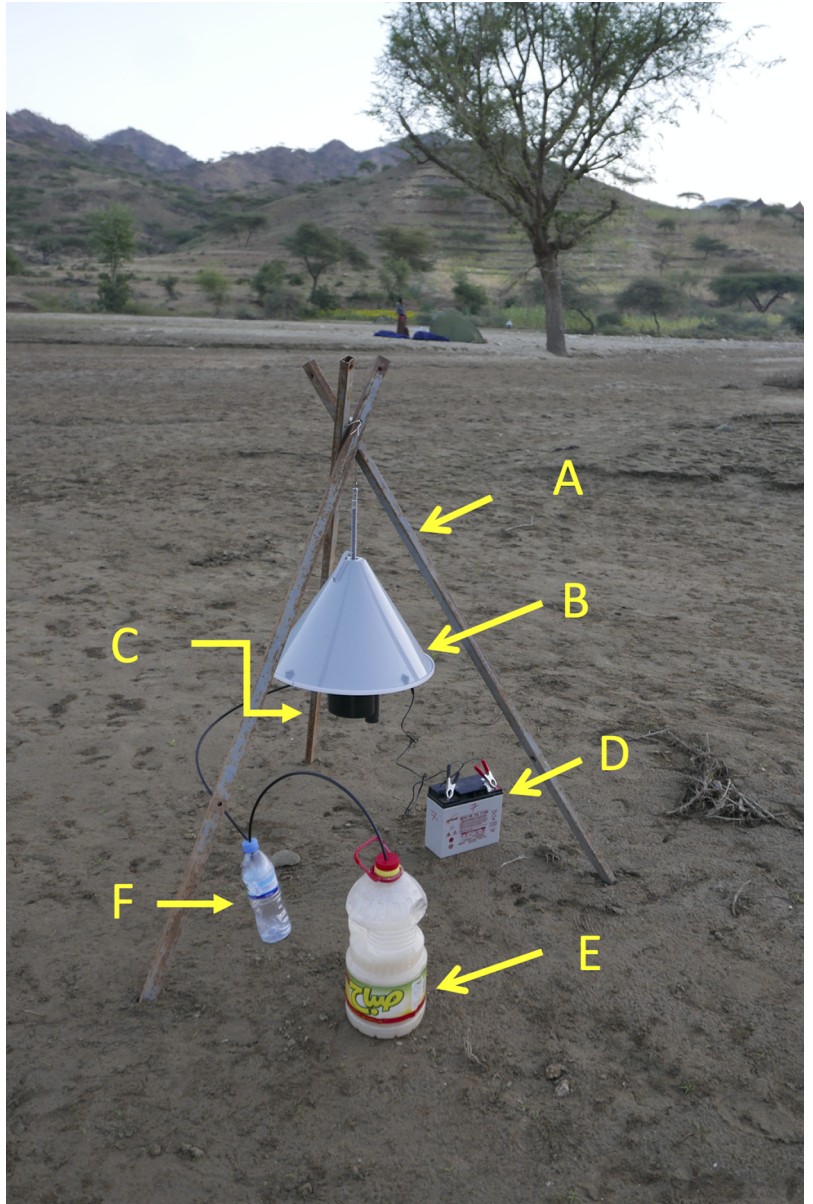

**Figure 4 Suna trap with carbon dioxide.** Suna trap in situ in an area close to Adi Boskal. (A) Tripod supporting the trap, (B) trap cover, (C) trap inlet, (D) 12 volt lead/acid battery powering the trap, (E) 5 litre bottle containing the sugar/yeast mix used to generate carbon dioxide, and (F) overflow bottle. An artificial lure is also placed inside the trap. Note the tent-trap in the distance. Photo © JD Charlwood.

*An. gambiae* complex mosquitoes collected were identified to species by PCR using the techniques of *Scott, Brogdon & Collins (1993)*.

### Resting collections

In order to determine the duration of resting inside the sheds after feeding, the proportion of gravid: blood-fed females on days following three or more sequential days of prior collections was compared to days when the sheds had been left undisturbed for a day

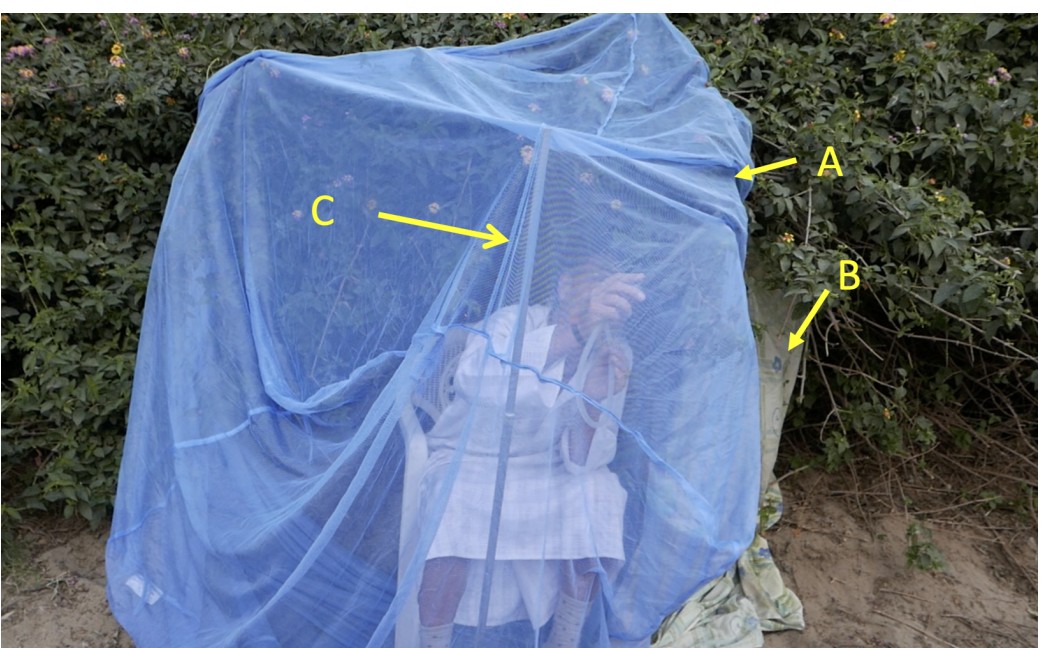

**Figure 5** **Set up of mosquito net over vegetation for use in the collection of, mosquitoes leaving diurnal resting at dusk.** (A) Double sized mosquito net placed horizontally over opening in vegetation, (B) material used to cover any excess openings, (C) pole to support net in place.

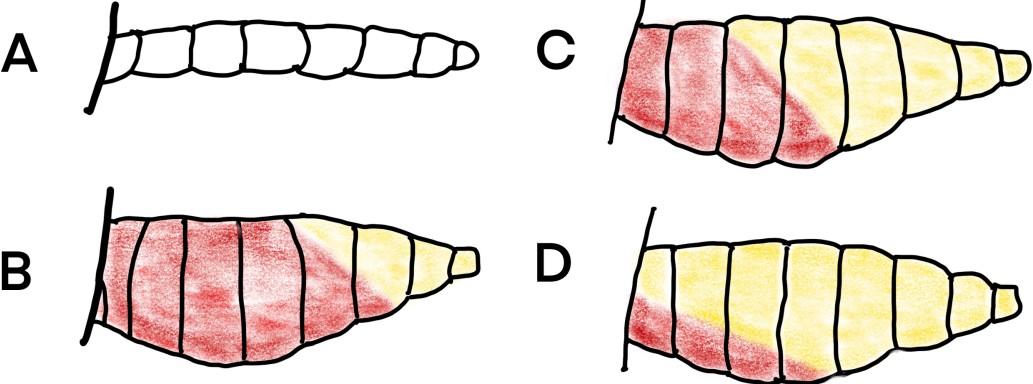

**Figure 6** **Daigram of mosquito abdomens at different stages of gonotrophic development.** (A) unfed, no blood or developing eggs visible; (B) engorged, abdomen full of fresh blood; (C) semi-gravid, anterior half of abdomen with dark blood and posterior half showing developing eggs; and (D) gravid, abdomen full of eggs (Redrawn from *Gillies et al., 1961*).

previously. *Anopheles* are generally gonotrophically concordant and each blood meal gives rise to complete egg batch. Should blood-fed insects have remained in the shed then they would have become semi-gravid or gravid by the second day, in which case the ratio of semi-gravid and gravid insects to the rest of the population would increase; but if they normally left shortly after feeding then the proportion of semi-gravid and gravid insects would not increase despite this respite in sampling.

*Anopheles arabiensis* may also take a pre-gravid blood meal during the first oviposition cycle that is used for adult nutrition rather than egg development. By maintaining blood-fed females alive for 24 h it is possible to separate those females that will develop eggs and will become gravid and those that will not develop eggs, despite being blood fed on collection. These females are pre-gravid and are almost entirely newly emerged (*Gillies, 1955*). The number of such pre-gravid females is, therefore, one indicator of adult recruitment to the population, and their relative frequency an indicator of mosquito survival rates (*Charlwood et al., 2003*). On 6 days engorged mosquitoes collected resting were kept in the insectary for 24 h and the proportion pre-gravid (i.e. the proportion that did not develop eggs) determined. Results were compared to estimates obtained by dissection of host-seeking mosquitoes. In this case virgin females, with undeveloped ovaries, that attempt to feed, will not develop eggs and so would go through a pre-gravid phase.

In order to determine post-prandial behaviour a series of three capture-recapture experiments using engorged mosquitoes from resting collections were performed. Engorged insects, collected resting from shed 1, were counted, dusted with fluorescent powder and released back into the shed. Mosquitoes from all subsequent collections (resting, host-seeking and exiting vegetation) were scanned with UV-light for colour which revealed previous marking.

### Host seeking collections

Mosquitoes from the tent-trap collections were dissected for parity, sac-stage and mated status according to the schema outlined in *Charlwood et al. (2003)*.

### Statistical analysis

Mosquito density by trapping method, period and shed was expressed in terms of Williams geometric mean ($M_w$). Williams means have been shown to be a robust measure of central tendency compared to arithmetic means, as they are less sensitive to both abundance and periodicity of insect occurrence (*Williams, 1937*, *Haddow, 1954*). The absolute density of mosquitoes in each shed was estimated using the maximum weighted likelihood method of *Zippin (1956*, *1958)* as modified by *Carle & Strub (1978)*. In this case the population size, $N$, is estimated as the smallest integer greater than the total catch, $T$, that satisfies the following inequality: $\left(\frac{N+1}{N-T+1}\right)\left(\frac{kN-M+T+0.5K}{kN-M+1+0.5K}\right)^k \leq 1$, where $k$ is the total number of removal periods. The parameter $M$ is estimated by the equation $M = \sum_{i=1}^{k}(k-i)c_i$, where $c_i$ is the number of individuals sampled.

Generalized Linear Multilevel Models (GLMMs) with negative binomial error distribution and log-link functions were applied to model the difference in sampling catches produced by the three resting collection methods, that is, Manual Aspirator, Backpack and Procopack. Collection method (a three levels factor: Manual Aspirator, Backpack and Procopack), shed (a binary factor: Shed 1 and Shed 2) and collection technique (a five-level factor: 1st–5th rounds) were considered as fixed factors. To accommodate potential hierarchical (correlation) and explicit nesting structure of repeated measurements across collectors and during the study period, as evidenced by exploratory

analysis, both collector and time (in days) effects were modelled as crossed random factors. This allowed the performance of each collection method to vary among collectors and days within sheds. Model fit was assessed by visual inspection of the graph of standardized model residuals against fitted observations (*Hartig, 2020*). The amount of variation of mosquito counts explained by predictors was determined by both marginal and conditional coefficients of determination $R^2$, proposed in *Nakagawa & Schielzeth (2013)*. Additionally, Akaike Information Criterion test (AICc) was applied to guide the selection of the best model fit. The most parsimonious model (the one with the lowest AICc) was preferred among the others. Final estimation of the mean and 95% Credible Interval bands (95% CrI) of fixed effect coefficients were obtained via simulations from posterior distribution of best model fit parameters. A total of 2,000 random simulations were performed using the function sim of the arm packages (*Gelman & Yu-Sung, 2018*). The performance of the CDC Backpack Aspirator and Prokopack in relation to Manual Aspiration was estimated in terms of Incidence Rate Ratio (IRR). The Tukey *post-hoc* multiple comparisons test was applied to determine the significance of the difference between main effect treatment levels and interacting effects levels. GLMM and *post-hoc* tests were performed using the packages lme4 v.1.1-23 [40] and emmeans v.1.4.8 (*Lenth, 2020*), respectively. Similarly, GLMMs was also applied to investigate the significance of the difference between mosquito catches obtained by Tent-traps and Suna trap. All the data processing tasks and statistical analysis were performed using the R software version 4.0.2 (*R Core Team, 2020*).

Ethical approval for the study was provided by the Research Ethical Clearance Committee of the Asmara College of Health Sciences on the 18/09/2017.

## RESULTS

Temperatures during the night dropped to a minimum of 11.8 °C outdoors but were 4 °C warmer inside the tents (minimum of 15.1 °C inside the tents). All 37 *An. gambiae* s.l. from resting collections identified to species by PCR were *An. arabiensis*. Since this is the only member of the species complex to have ever been found in Eritrea (*Shililu et al., 2004*) we assume that this was the only species collected. A total of 5,382 host seeking, 2,296 resting and 357 *An. arabiensis* exiting vegetation were collected during the experiment. In addition, 1 male and 1 female *An. demellioni*, 6 *An. garnhami* and 1 *An. turkhudi* were collected resting towards the end of the study.

Greatest numbers of host-seeking females were collected in the sentinel tent-trap run for 23 consecutive nights during the experiment (geometric mean of 162 per night 95% CI [117.13–224.87]). The majority of host-seeking parous mosquitoes had large follicular sacs, indicating a rapid return to host-seeking after oviposition (*Charlwood et al., 2018a*). Total numbers collected declined during the experiment whilst at the same time the parous rate among the 281 mosquitoes dissected (*Charlwood et al., 2018a*) increased from 21% to 56% (Fig. 7). Thus, there was a decline in output of newly emerged insects and the population was an ageing one. Collections from 49 tent-traps, eight all night landing collections and 28 resting collections were positively correlated (Pearson's $r = 0.64$ $n = 5$, n. s, between tent and landing collections; $r = 0.76$, $n = 5$, $P = 0.05$ between tent and

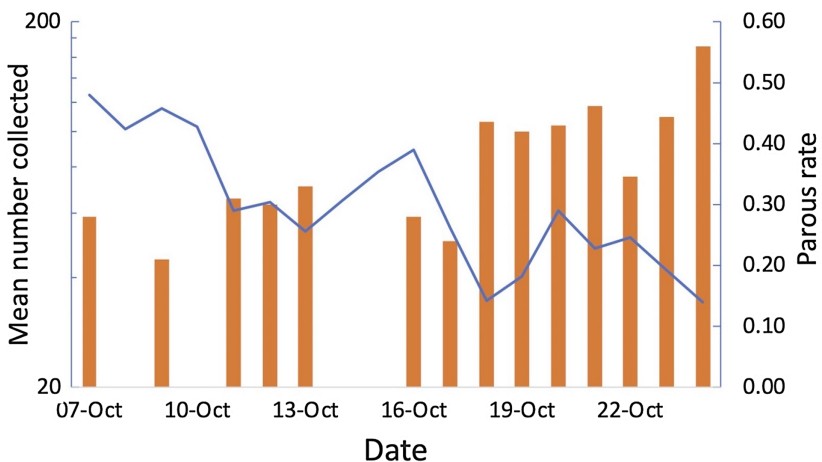

**Figure 7 Tent traps and parous rates.** Mean number of *Anopheles arabiensis* collected in experimental tent traps (on a log scale) and proportion of the collection that was parous by date of collection, Adi Boskal, Anseba zoba, Eritrea.

**Table 1 Comparison of abundance of *An. arabiensis* found resting indoors at site 1 and site 2 estimated by removal sampling. N: Estimated total abundance and *p*: probability of mosquito collection.**

| Method | Parameter | Shed 1 | SE | 95% CI | Proportion in round 1 | Shed 2 | SE | 95% CI | Proportion in round 1 |
|---|---|---|---|---|---|---|---|---|---|
| Manual Aspirator | Total | 100 | | | 0.34 | 165 | | | 0.25 |
| | Estimate | 207 | 4.42 | [198.33–215.67] | | 218 | 4.77 | [208.65–227.35] | |
| | *p* | 0.46 | 0.03 | [0.39–0.53] | | 0.45 | 0.03 | [0.39–0.52] | |
| Backpack | Total | 382 | | | 0.67 | 56 | | | 0.34 |
| | Estimate | 725 | 1.78 | [721.52–728.48] | | 161 | 2.85 | [155.41–166.59] | |
| | *p* | 0.67 | 0.02 | [0.64–0.70] | | 0.51 | 0.04 | [0.43–0.58] | |
| Procopack | Total | 318 | | | 0.54 | 302 | | | 0.59 |
| | Estimate | 648 | 3.43 | [641.27–654.73] | | 371 | 1.27 | [368.51–373.49] | |
| | *p* | 0.58 | 0.02 | [0.55–0.61] | | 0.67 | 0.02 | [0.63–0.71] | |

resting collections; $r = 0.97$, $n = 12$, $P = >0.01$ between landing and resting collections). The data from all collections is provided in the Supplemental File 3.

## Resting collections
### *Mosquito counts by collection method*
Shed 1 was darker than shed 2 (20 vs 40 Lux measured at 09:00). Temperature extremes were also greater in the more open Shed 2 (maximum and minimum temperatures recorded close to the roof of the sheds over 24 h being 46.7 °C and 18.2 °C in Shed 2 compared to 39.6 °C and 17.2 °C in Shed 1). A total of 1,561 and 735 female and 93 and 19 male *An. arabiensis* were collected from Shed 1 and Shed 2, respectively from 134 rounds of sampling. The estimated abundance of mosquitoes in the two sheds according to the different methods of collection with their 95% confidence intervals are shown in Table 1.

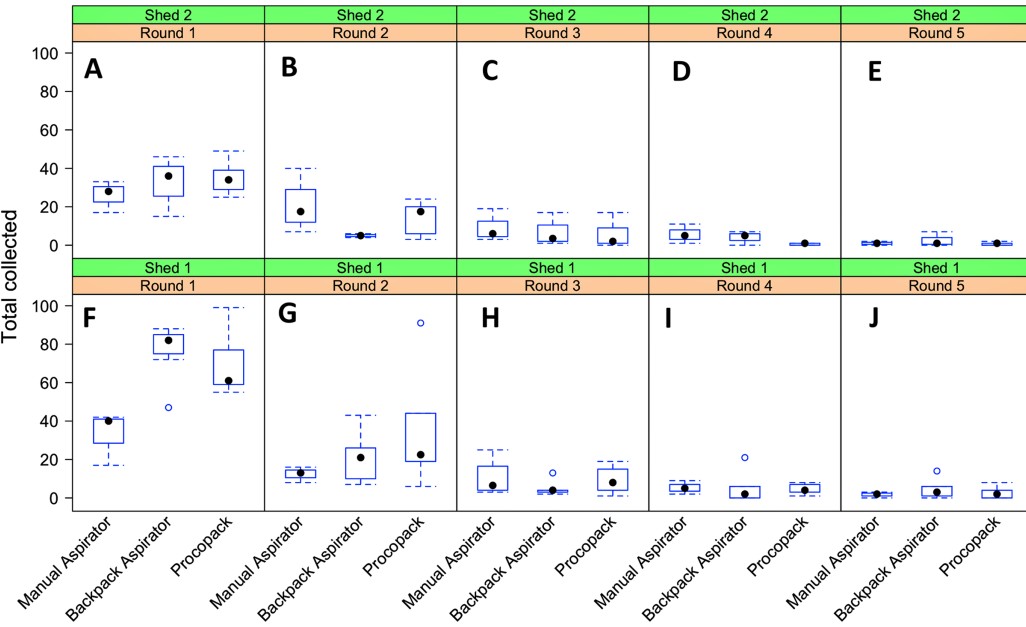

**Figure 8 Resting collections of *Anopheles arabiensis*.** Relationship between collection methods, shed and sampling round on numbers of *Anopheles arabiensis* caught resting, Adi Boskal, Eritrea. (A) Shed 2, Round 1. (B) Shed 2, Round 2. (C) Shed 2 Round 3. (D) Shed 2 Round 4. (E) Shed 2 Round 5. (F) Shed 1 Round 1. (G) Shed 1 Round 2. (H) Shed 1 Round 3. (I) Shed 1 Round 4. (J) Shed 1 Round 5. In each box the method used from left to right was Manual aspirator, Backpack aspirator and Procopack aspirator.

Relatively high mosquito counts were obtained from Shed 1 using either the Backpack or Procopack compared to manual aspiration (Table 1). Manual aspiration produced a total of 198 mosquitoes (Shed 1) and 208 mosquitoes (Shed 2), Backpack (723 vs. 157) and Procopack (640 vs. 370). Results of estimation of total abundance of indoor resting mosquito population by removal test (Table 1) suggest that manual aspiration underestimated density in both sheds. On the other hand, the Backpack may have underestimated the population abundance at Shed 2 compared to manual aspirator and, especially, Procopack. The overall estimate of total abundance (± 95% CI) by combining catches from all the three methods were 1580 (1567–1585) and 750 (738–754) from shed 1 and shed 2, respectively. The probability of capture was 0.603 (0.581–0.624) and 0.566 (0.534–0.598).

### Sampling regime and collection performance

There was a sharp reduction in the number of mosquitoes collected after the first sampling round, irrespective of collection method or shed but the three methods produced similar numbers caught after the second round of collection (Fig. 8). The distribution of mosquito counts by the three methods were heavily (right) skewed (Supplemental File 4). Further exploratory analysis suggested that the counts could plausibly be described by a negative binomial distribution function. Additionally, since collections were repeatedly performed by the same individuals at the same sheds (shed 1 and shed 2) over the study period some amount of dependency (correlation structure) between observations was introduced (Supplemental File 4).

The best model fit of sampling method performance (AICc = 852.2) was one that included collection method and shed as main effects factors and an interaction effect between collection methods and sampling regime. Collectors and day of collection were considered as random factors. Both fixed and random factors explained 86.6% of the variability in mosquito counts (Fig. 9). Results of mosquito Incidence Rate Risks (IRR ± 95CrI) indicate that a mosquito was, respectively, 1.45 (0.89–2.32) and 1.83 (1.17–2.29) more likely to be sampled by CDC Backpack and Prokopack compared to manual aspiration. However, the difference was statistically significant only between manual aspiration vs Prokopack. The results also showed that the likelihood of collecting any mosquito varied significantly between sheds; that is, the chance of collecting at Shed 2 was 41.6% lower compared to Shed 1. There were also significant interaction effects between collection methods and sampling regimes.

There was a significant difference between the relative proportions of *An. arabiensis* females in different abdominal stages collected from shed 1 and shed 2 on the days when no collections were conducted the previous day ($X^2$ with Yates correction 23. 5, $p < 0.00002$). (Table 2) . On the 10 days when resting collections had been conducted the previous day the relative proportions of *An. arabiensis* by abdominal stage from either shed were similar ($X^2$ with Yates correction = 2.3963, $p = 0.12$) but the proportion of semi-gravid and gravid *An. arabiensis* females collected from Shed 1 were significantly higher than Shed 2, the lighter shed, following days in which no previous collections had been undertaken ($X^2$ with Yates correction is 7.3493, $p = 0.0067$). The number of females collected from Shed 2 was not significantly different from numbers collected following days where previous collections had been undertaken (DRR = 0.84, 95% CI [0.56–1.26], $P = 0.41$).

### Pre-gravid rates

Seventy-two (35%) of the 204 engorged mosquitoes collected resting and kept in the insectary for 24 h failed to develop eggs, and so were pre-gravid. This proportion was similar to the 86 (31%) of the 281 dissected insects that were virgins ($X^2 = 1.6$, $P = 0.21$). Fifty-one (37%) of the recently emerged insects dissected had a mating plug. These insects may not develop eggs after a full blood meal and implies that at least two thirds of newly emerged insects went through a pre-gravid stage.

### Exit collections

Males started to exit the vegetation when light levels fell to approximately 500 Lux each day (mean = 506 Lux; standard deviation = 80 Lux; range 373–610 Lux). Peak exit rates of males occurred 3 min after the initial mosquito was collected. Both unfed and gravid females were collected leaving the vegetation a few minutes after the males (Fig. 10). One of the 13 gravid females collected had been and marked 2 days earlier and released, engorged, back into Shed 1.

### Performance of host-seeking sampling methods

The number collected in the HLC decreased with time during the night as did the temperature. The geometric mean number per night in the HLC was 35.02 (s.d. 12.9). After

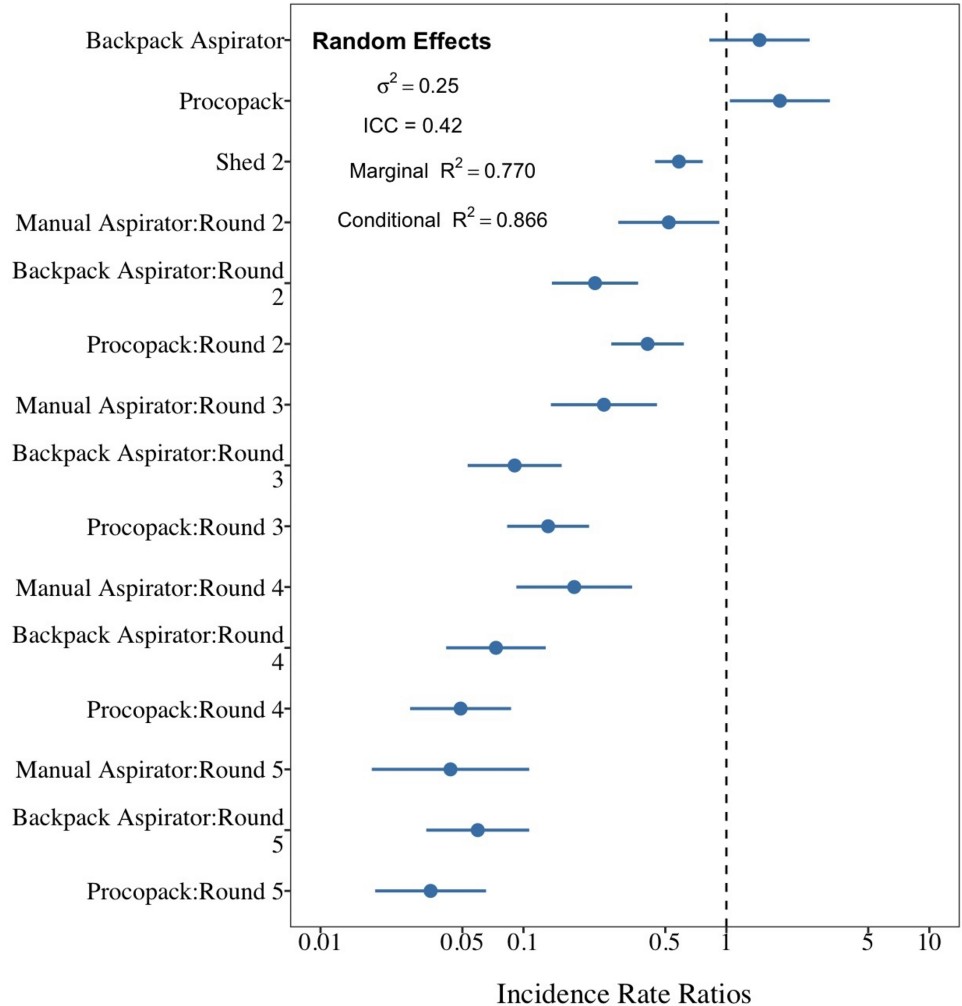

**Figure 9 Comparison between sampling techniques for indoor resting mosquitoes.** Performance of Backpack and Prokopack sampling indoor resting *Anopheles arabiensis* compared to manual aspiration (reference), Adi Boskal, Eritrea.

eight nights of collection a total of 40 *An. arabiensis* (mean of 5 per night 95% CI [1.18–8.82]) were collected in the Suna trap with carbon dioxide and only 10 were collected when the trap was used without carbon dioxide (mean 1.25 per night, 95% CI [0–3.6]) (Table 3). The Suna trap also failed to collect any mosquitoes when it (with added carbon dioxide) was moved alongside a likely flight path of the mosquitoes by the vegetation 15 m from the sentinel tent-trap, or close to the base of the hill leading to the village. Furvela traps type I and type II collected, respectively, 6.23 (4.90–7.97) and 4.63 (3.42–7.96) significantly more mosquitoes than did the HLC. Both types of Furvela tent traps collected relatively similar numbers of mosquito (Incidence Ratio = 1.36; 95% CI [0.917–2.013]) (Fig. 11).

A previously marked blood-fed mosquito, was recaptured (with notably large follicular sacs), host-seeking in the sentinel tent-trap, 2 days after release in Shed 1.

**Table 2 Mean number of *An. arabiensis* collected from two animal sheds when collections had or had not been made the previous day, Adi Boskal, Anseba zoba, Eritrea (UF, unfed; PF, part-fed; BF, blood-fed; SG, semi-gravid; GR, gravid).**

|  | UF % [95% CI] | PF % [95% CI] | BF % [95% CI] | SG % [95% CI] | GR % [95% CI] |
|---|---|---|---|---|---|
| No previous collection |  |  |  |  |  |
| Total Shed 1 | 4 | 26 | 189 | 80 | 145 |
| Total Shed 2 | 10 | 35 | 126 | 44 | 34 |
| Shed 1 | 0.9 [0.02–1.8] | 5.9 [3.7–8.0] | 42.6 [38.0–47.2] | 18.0 [14.4–21.6] | 32.7 [28.3–37.0] |
| Shed 2 | 4.0 [1.6–6.5] | 14.1 [9.7–18.4] | 50.6 [44.4–56.8] | 17.7 [12.9–22.4] | 13.7 [9.4–17.9] |
| Previous collection |  |  |  |  |  |
| Total Shed 1 | 19 | 36 | 167 | 79 | 55 |
| Total Shed 2 | 15 | 42 | 131 | 41 | 45 |
| Shed 1 | 5.3 [3.0–7.7] | 10.1 [7.0–13.2] | 46.9 [41.7–52.1] | 22.2 [17.9–26.5] | 15.5 [11.7–19.2] |
| Shed 2 | 5.5 [2.8–8.2] | 15.3 [11.1–19.6] | 47.8 [41.9–53.7] | 15.0 [10.7–19.2] | 16.4 [12.0–17.9] |

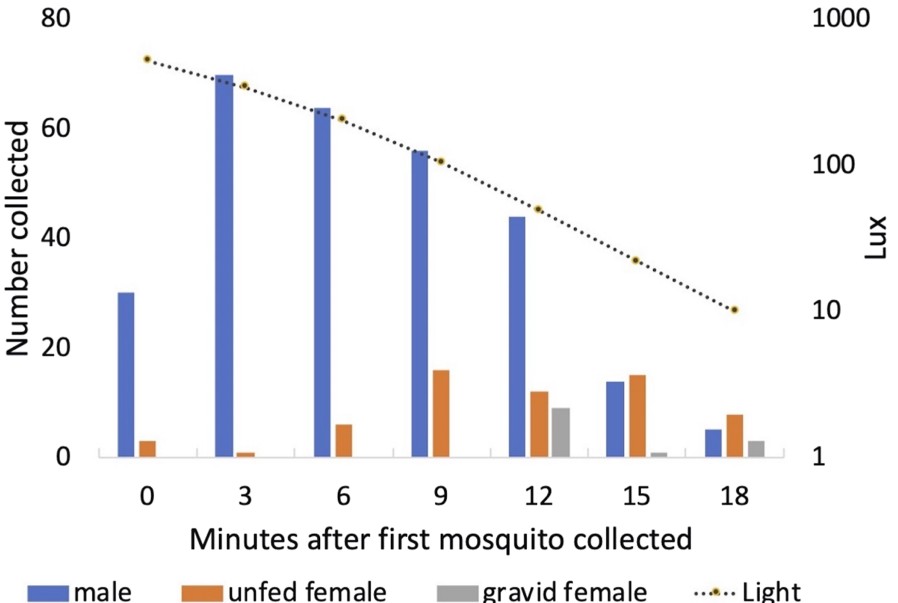

**Figure 10 Collection of mosquitoes exiting vegetation at sunset.** Numbers of male, unfed female and gravid female *Anopheles arabiensis* exiting vegetation according to light-level, Adi Boskal, Eritrea.

The addition of the lure to the tent-trap, either in the sentinel trap or the experimental tents, did not result in an increase in the number of mosquitoes collected ($t^7 = 0.28$, $P = 0.79$ for sentinel tent trap; $t^7 = 0.48$, $P = 0.64$ for experimental tents).

## DISCUSSION

Monitoring outdoor biting and resting malaria vectors has assumed a greater importance as a result of worldwide efforts to eliminate the disease (*World Health Organization, 2018*). Previous work has highlighted the potential importance of outdoor exposure to *An.*

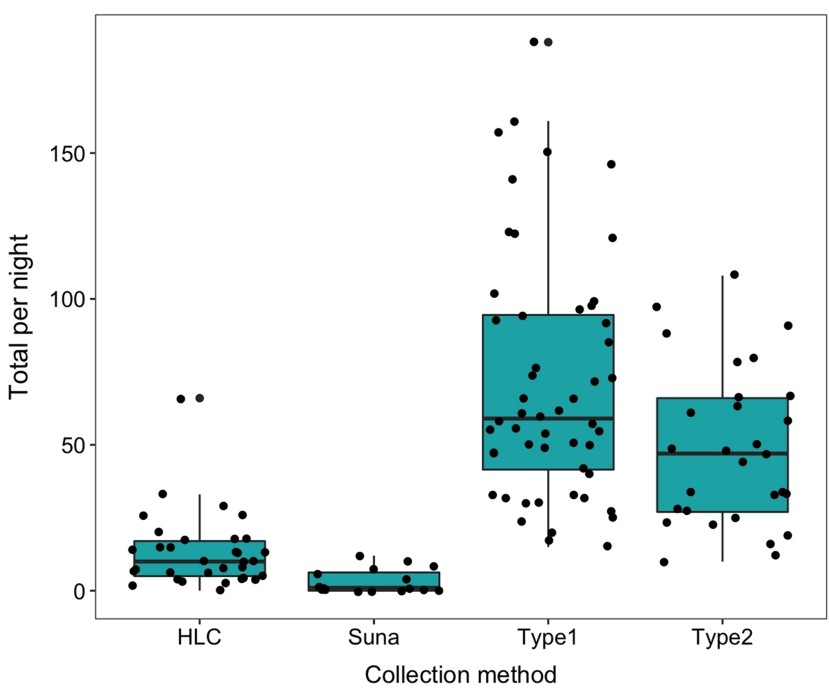

**Figure 11 Comparison between collection methods of host seeking mosquitoes.** Density of host-seeking *Anopheles arabiensis* obtained by different sampling methods in Adi Boskal, Eritrea.

**Table 3 Number of collections, total collected geometric mean number per collection and standard deviation of *An. arabiensis* collected host seeking, Adi Boskal, Anseba zoba, Eritrea.**

| Collection | HLC | Tent | Sentinel | Suna with $CO_2$ | Suna without $CO_2$ |
|---|---|---|---|---|---|
| *n* | 8 | 19 | 30 | 8 | 8 |
| Total collected | 296 | 1,248 | 2,107 | 40 | 10 |
| Geometric Mean | 35.0 | 72.4 | 162.2 | 3.1 | 0.6 |
| S.D. | 12.9 | 19.3 | 45.7 | 4.6 | 2.7 |

*arabiensis* in Eritrea (*Shililu et al., 2004*). Two alternatives were tested to HLC for the monitoring of outdoor biting mosquitoes in Adi Boskal, the Suna trap and the Furvela tent-trap. In comparisons between an earlier version of the Furvela trap and HLC, where *An. gambiae* was the principle vector collected, the methods were similar at low densities but at higher densities HLC was considered to be more efficient (*Govella et al., 2009*). Nevertheless, several thousand mosquitoes have been caught in a night with a single trap (*Charlwood et al., 2011*, *2012*, *2013*).

The Suna trap caught very small numbers of *An. arabiensis* compared to the other collection methods. *Cribellier et al. (2020)* have recently shown that adding heat to a trap increases its efficiency. Thus, one reason why the Suna trap failed to collect mosquitoes may be because the trap did not have a sufficient 'heat signature' that might otherwise aid mosquitoes in location when in close proximity to potential hosts. It is also possible

that the lure used failed to elicit a response from the mosquito. The fact that numbers were not affected in the sentinel trap when a lure was attached to the opening and the relative inefficiency of the trap against *An. gambiae* s.l. in Kenya (*Homan et al., 2016*) and Tanzania (*Cribellier et al., 2020*) indicates that this might be the case. On the other hand, our results indicate that Furvela tent-traps are a suitable, indeed a more efficient, alternative to HLC for the collection of outdoor biting *An. arabiensis* in this area. At the temperatures experienced, collectors' legs became cold and may also have not given off heat cues that assist host location. It is therefore possible that the difference between tent-trap collections and HLC may be less apparent at warmer temperatures. Nevertheless, there remain considerable advantages in using tent-traps, not the least of which are getting a good night's sleep, not being exposed to pathogens and having a uniform collection efficiency between tents. The method used to attach the CDC-trap to the tent, however, did not significantly affect the number of *An. arabiensis* collected.

In the present experiments the sentinel trap caught more than twice as many mosquitoes as the other tents. There are a number of possible reasons for this: the odour of the host in the tent (JDC) whilst being unappealing to humans may have been particularly attractive to mosquitoes; the tent was larger than the other tents and the enhanced visual profile may have attracted more mosquitoes (*Hawkes & Gibson, 2016*). Because mosquitoes tend to fly along the edge of vegetation (*Charlwood & Wilkes, 1981*) those that rested in vegetation close to the river, or that oviposited prior to host seeking may have been funnelled through the opening where the sentinel trap was located. Thus, the trap may have sampled from a more concentrated population than the other traps and so caught more mosquitoes as a result. Setting up tent-traps in such locations may enhance collections, which is especially useful at low mosquito population densities.

The collection of resting females also forms an important component of vector monitoring. The results of sequential sampling using either the CDC-backpack aspirator, a Prokopack aspirator, or manual aspiration were equivocal. For removal sampling to function adequately a number of assumptions must be met: the catching procedure must not affect the probability of an animal being caught; the population must remain stable during the catching period, sampling effort must be the same in each round of collection and, most importantly, the chance of being caught must be equal for all animals. A relatively large proportion of the population must also be caught to obtain reasonably precise estimates. Numbers collected on each trapping interval must also decline for estimates to be meaningful (*Southwood, 1978*). Whilst we were unable to estimate total numbers in the sheds on all but a few occasions our results indicate that between a quarter and two thirds of the total sample was obtained during the initial round of collection depending on the shed and method used (Table 1). Thus, a single round of collection from an indoor resting site is a suitable sampling procedure. It being better to sample five sheds, once than one shed five times.

The number of mosquitoes collected by manual aspiration, especially in Shed 1 was lower than the other methods. This is probably because not only was the shed dark but there were many hard to see hiding places due to the shed walls being made of stones. It was not easy to illuminate these hiding places with a torch. These considerations did not

affect collection by mechanical aspirator. In Tanzania, *Maia et al. (2011)* found that CDC backpack and Prokopack aspirators were equivalent in efficiency for collecting mosquitoes in general, but that the Prokopack was easier to use and there was increased consistency across the numbers of mosquitoes collected by four different collectors operating the Prokopack compared with the CDC-BP. Also, in Tanzania the Prokopack was considered to be a better method for collection of resting insects than manual aspiration (*Charlwood et al., 2018b*). It is now the method of choice for such samples.

The blood-fed *An. arabiensis* released in the animal shed in the middle of the village and recaptured gravid 2 days later exiting vegetation close to the oviposition site at dusk indicates that females probably arrive there half-gravid. Seeking a resting site close to the oviposition site when half gravid makes sense if the oviposition site is some distance from the (indoor) feeding site. It is undertaken when the insect still may have the ability to use energy from the blood meal for flight (although by being outside nectar sources may be available) and means that the insects can start hunting early in the night and shortly after oviposition. They will not have used energy unnecessarily flying from the village to the oviposition site before starting to hunt, which they did shortly after oviposition, as shown by the high proportion of dissected females with large follicular sacs (*Charlwood et al., 2018a*), including one marked and released in the same shed 2 days previously. The marked mosquito collected from the tent-trap had presumably fed on a non-human host prior to release. These results emphasise the rapid gonotrophic cycle in this species and also suggests a lack of 'host fidelity' in the mosquito.

Male *arabiensis* in Adi Boskal appeared to spend much of their lives in vegetation close to the emergence site. It was from here that they were collected at dusk. Emergence from vegetation was, as expected, related to illumination and, as expected, males left before the females. Most of the females collected exiting vegetation were unfed. Such females are largely newly emerged virgins who would presumably mate in swarms, which were seen at this time in the river bed, before ascending the hill to the village for their first blood meal. How, or where, pre-gravid virgin females, who feed in the village, mate remains unknown. Less than 5% of the insects collected from the animal sheds were males and alternative sites for males to rest were not obvious. Unfortunately searching for and monitoring of swarms in the village was not undertaken. Similarly, the question of how the mosquito (and the other anophelines collected) maintain themselves at times of low population density is not known and whether they do form part of a meta-population (which implies occasional extinction) remains moot.

The likelihood that blood-fed insects will complete gonotrophic development inside human-made structures appeared to depend on illumination, the darker of the two sheds in Adi Boskal contained a higher proportion of gravid insects on the days when the sheds had not been sampled the previous day compared to the lighter shed. The absence of any difference between days when sampling had been curtailed on the previous day in the lighter shed indicates that the insects were leaving before they became gravid. This implies that the insects had left the lighter shed to rest outside. Resting outside shortly after feeding may have a significant mortality risk for mosquitoes (and may be one reason why species that rest inside houses are better vectors than those that rest outdoors) (*Gillies,*

*1954*). Constructing well-lit, rather than dark, animal sheds, may encourage otherwise endophilic mosquitoes to leave and so reduce their survival and hence reduce malaria transmission.

Given that the population was not stable during the study, but rather suffered a decline in output, estimation of survival by parous rate determination was not possible. The data was also insufficient to determine survival rates by time-series analysis. Nevertheless, just over a third of all the insects from the resting collections were pre-gravid whilst almost half the mosquitoes collected from the tent-trap had stage I ovaries, suggesting that they were also pre-gravid females. Thus, newly emerged mosquitoes comprised a large proportion of the collections, indicating a relatively low survival rate in the mosquito. The higher proportion of mosquitoes that were considered to be pre-gravid by dissection compared to those from resting collections may reflect mortality amongst such females before they could obtain a blood meal (as appears to happen with young *An. coluzzii* from São Tomé (*Charlwood et al., 2003*)). All older females were gonotrophically concordant and took 2/3 days per oviposition cycle.

Owing to the low night-time temperatures in Adi Boskal, people enter their houses early in the evening. Making sure that houses are mosquito-proof and that villagers have access to LLINs should help protect them from malaria. Whether anything other than current control techniques (LLINs over all beds in the village) needs to be used in Adi Boskal is moot. There had been no cases of malaria reported from Adi Boskal in the previous year and so the village would seem to be a case of *anophelism sans malaria*. Although it is not certain that the mosquito provides any benefit to local ecosystems it may do so. Treating cattle or spraying the shed with insecticide would also, perhaps, enhance the development of resistance in the mosquito and waste precious resources.

## CONCLUSIONS

At the time of the study, although it would readily bite humans if they were available, the *Anopheles arabiensis* population from Adi Boskal was apparently a largely zoophagic one. Most of the newly emerged female mosquitoes went through a pre-gravid phase after which they became gonotrophically concordant with a rapid gonotrophic cycle. A proportion of the semi-gravid and gravid insects would leave their resting site to rest in vegetation close to the stream below the village prior to oviposition. Male mosquitoes rested in vegetation and swarmed close to the stream below the village. Furvela tent-traps set up in areas where there may be a flyway. are a suitable alternative to human landing catches for the collection of outdoor biting *An. arabiensis* whilst the Prokopack mechanical aspirator is the method of choice for the collection of resting insects. Constructing well-lit, rather than dark, animal sheds, may encourage otherwise endophilic mosquitoes to leave and so reduce their survival and hence their vectorial capacity.

## ABBREVIATIONS

| | |
|---|---|
| **CDC** | Centres for Disease Control |
| **HLC** | Human Landing Collection |
| **IRS** | Indoor Residual Spray |

**LLIN**        Long Lasting Insecticide-treated Net
**WHO**        World Health Organization

## ACKNOWLEDGEMENTS

We would like to thank the students of the College of Health Sciences, especially Kokob Tesfay, for their assistance during the experiments. We would also like to thank the editor and reviewers for their perceptive comments which helped us to considerably improve the manuscript.

### Funding

There was no funding for this work.

### Competing Interests

The authors declare that they have no competing interests.

### Author Contributions

- Jacques D. Charlwood conceived and designed the experiments, performed the experiments, analyzed the data, prepared figures and/or tables, authored or reviewed drafts of the paper, and approved the final draft.
- Amanuel Kidane Andegiorgish conceived and designed the experiments, authored or reviewed drafts of the paper, helped set up the fieldwork and provided logistical support, and approved the final draft.
- Yonatan Estifanos Asfaha conceived and designed the experiments, performed the experiments, authored or reviewed drafts of the paper, and approved the final draft.
- Liya Tekle Weldu analyzed the data, authored or reviewed drafts of the paper, and approved the final draft.
- Feven Petros conceived and designed the experiments, performed the experiments, authored or reviewed drafts of the paper, and approved the final draft.
- Lidia Legese performed the experiments, authored or reviewed drafts of the paper, and approved the final draft.
- Robel Afewerki analyzed the data, authored or reviewed drafts of the paper, and approved the final draft.
- Selam Mihreteab conceived and designed the experiments, authored or reviewed drafts of the paper, helped set up the fieldwork and provided logistical support, and approved the final draft.
- Corey LeClair conceived and designed the experiments, analyzed the data, authored or reviewed drafts of the paper, and approved the final draft.
- Ayubo Kampango analyzed the data, prepared figures and/or tables, authored or reviewed drafts of the paper, and approved the final draft.

## Animal Ethics

The following information was supplied relating to ethical approvals (i.e., approving body and any reference numbers):

The Research Ethical Clearance Committee of Asmara College of Health Sciences approved this research project.

## Field Study Permissions

The following information was supplied relating to field study approvals (i.e., approving body and any reference numbers):

A field permit was not required. The work was undertaken at the same site that a previous PeerJ article 'We like it wet' - by the same authors - was undertaken.

## Data Availability

Raw data and the statistical analysis in greater detail is available in the Supplemental Files.

## Supplemental Information

Supplemental information for this article can be found online at http://dx.doi.org/10.7717/peerj.11497#supplemental-information.

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
