# Peer review of "Novel sampling methods for monitoring Anopheles arabiensis from Eritrea"

_PeerJ, doi:10.7717/peerj.11497_

## Round 0.1 · original submission · Major Revisions

Dear Authors,

Please answer the reviewers' comments and modify the manuscript accordingly. Both the reviewers are positive towards your manuscript but have some comments.

Sincerely,

Gunjan

·

Basic reporting

See general comments

Experimental design

See general comments

Validity of the findings

See general comments

Additional comments

This is an interesting manuscript that address a topic of unquestionable public health relevance. The authors present a complete study that aims at evaluating different trapping methods for host-seeking and resting mosquitoes. Being able to assess local mosquito populations is important to apply adapted control measures. Moreover, sampling outdoor biting mosquitoes is essential if malaria elimination is aimed.
On the other hand, multiple methods are presented, and the understanding of the methodology could be complex for the reader. Moreover, some aspects of the manuscript and methodology are not clear. I recommend that the authors make the manuscript simpler by providing a clearer structure. In summary, I recommend the publication of the manuscript after some major revisions are implemented.

My comments:

Materials and Methods:
- It is not clear how many days the experiments last for. Could the authors explain how many times each of the collection methods were repeated?
- When explaining the methods, can the authors add subtitles indicating the methods used for resting, host-seeking and exiting vegetation mosquitoes?
- Authors use “shed”, “shelter” and “site” indistinctively. I think it will be easier to follow if they stick to using the same word along the manuscript.
- Could the authors provide a description of the terminology used to classify females according to their abdominal stage (including the definition of pre-gravid females)?
Line 119: Study design. Can the authors show in Fig.1 the disposition of the Latin square design experiment? Where is it located? In the “trapping sites”?
Line 131: Why were mosquitoes only collected in animal shelters and not in human dwellings? How many days were the collection techniques repeated?
Line 138: Can the authors be more specific about how the time sequence of the collections? (e.g. how many days they collected in a row? How many days there were no collections)
Line 181-185: These considerations could be moved into Discussion.
Line 204: Can the authors add a picture if the Suna Trap in the Figures?
Line 239: “females sorted according to their abdominal condition”. Please, include the terminology used.
Line 240: “species using the techniques of Scott et al”. Could the authors add that these are molecular techniques based in PCR. Which was the method used to collect these mosquitoes?
Line 259: It is not clear what the authors mean by “collection regimes”.
Results:
Line 297: “Numbers collected declined during the experiment whilst at”. I suggest “Total numbers collected declined…”
Line 311: A single quantity of males (112) is given, but there are 2 shelters?
Line 320: “the Backpack may have dramatically underestimated“. Backpack estimates 161, Manual does 218. Does not seem like a dramatic underestimation to me.
Line 352: “the proportion of blood-fed and gravid A. arabiensis females collected from Shed 1 were significantly higher than Shed 2”. I do not see this in Table 2 (Shed 1: 42.6%, Shed 2: 50.6%).
Line 371: Performance of host-seeking sampling methods. The authors provide different values in this paragraph (mosquitoes per night, total mosquito numbers, odds of collection) Could the authors be consistent in the values provided for the different methods (e.g. mosquitoes collected per night)?

Discussion: I miss the discussion of these results:
- Why is the Suna Trap not effective in this setting?
- Why more the greatest number of mosquitoes collected from the sentinel tent-trap?
- Why was the probability of mosquito catch by manual aspirator lower than the other methods?

Line 412: “Whilst we were unable to estimate total numbers in the sheds on all but a few occasions our results indicate that a single round of collection from an indoor resting site using a mechanical sampler rather than manual aspiration is a suitable sampling procedure”. Could the authors explained better how do they support this statement?

Figure 1:
I think this figure should include more information. What is the meaning of “outdoor trapping site”? Where are the animal shelters located? Where were the HLC performed?

Figure 4:
A legend is missing.

Figure 5: Could the authors use “shed” instead of “site”? Can authors name 1-5 as rounds?

Figure 6:
In the legend, what does “reference” refer to?

Figure 8: y-axis refers to total collected per night?

Table 1:
Could the authors include the real numbers of mosquitoes collected?

Table 2:
Legend: More than “mean numbers”, the table refers to “proportions”. Could the total numbers be shown?

Reviewer 2 ·

Basic reporting

Clarifcations needed & suggestions

Line 234 – Instead of a video a picture of the net can be given as a figure. If the video is to be included, it needs to be shortened.
Lines 310-311 & Lines 322-324 – Numbers given in Table 1 and those mentioned in the text do not tally
Line 348 – Change the first word ‘The’ to “There”
Lines 348-350 – If proportions for shed 1 and Shed 2 are given it would be better, or at least mention in which categories there was a difference- Table 2
Figure 3 – Picture showing the Furvela tent trap II may be deleted as there is a video showing the tent
Figure 4 - Caption says ‘A. arabiensis collected from experimental tents’ are these collections include only the two Furvela tent traps or sentinel tent trap and Suna trap as well?
Genus Anopheles is generally abbreviated as An.

Experimental design

No comment

Validity of the findings

No comment

---

## Round 0.2 · Major Revisions

I think the authors have done answered some of the comments. There are still many issues in the presentation. I think if authors cant have figures of the net then including a video is ok. Also, I think it will help readers if authors can provide a flow diagram of workflow and schematic diagram of the sampling method.

·

Basic reporting

see below

Experimental design

see below

Validity of the findings

see below

Additional comments

The track-changes word document that the authors have included is not the same one as the pdf. Unfortunately, I have used the word document for my review. There are some mistakes regarding the figure referencing in this word document. Please, make sure that this is not the case in the final manuscript. In general, I think the manuscript would benefit from a clearer structure that will facilitate the reading. In some parts, it takes time to understand which mosquito collection method the authors are referring to. I would recommend that the authors revise this one more to increase the understanding. Likewise, in some parts it is not clear how many time the experiments have been repeated.
Background:
Line 96: Perhaps the authors can indicate that collectors are used as baits in tent-traps. Otherwise it is difficult to follow the sentence flow.
Methods:
Line 140: The study design is still not completely cleared to me. The "Latin square design" rotated between the 5 positions, or was each sampling method in a different location each night? Please, clarify.
Line 218: Could the authors indicate that the Furvela tent-trap Type II corresponds to Fig. 4?
Line 252: Methods regarding swarm observations are not included
Line 261: I would add a new subsection regarding this paragraph, something like: “Mosquito classification”.
Figures showing different sampling methods will benefit from a better labelling to indicate the different trap components.

---

## Round 0.3 · accepted · Accept

Thanks for clarifying the reviewer's comments.